# MixLasso: Generalized Mixed Regression via Convex Atomic-Norm Regularization

**Ian E.H. Yen** [*†]    **Wei-Cheng Lee** [‡]    **Sung-En Chang** [‡]    **Kai Zhong** [§]
**Pradeep Ravikumar** [*]    **Shou-De Lin** [‡]

[*] Carnegie Mellon University    [†] Snap Inc.    [‡] National Taiwan University    [§] Amazon Inc.

## Abstract

We consider a generalization of mixed regression where the response is an additive combination of several mixture components. Standard mixed regression is a special case where each response is generated from exactly one component. Typical approaches to the mixture regression problem employ local search methods such as Expectation Maximization (EM) that are prone to spurious local optima. On the other hand, a number of recent theoretically-motivated *Tensor-based methods* either have high sample complexity, or require the knowledge of the input distribution, which is not available in most of practical situations. In this work, we study a novel convex estimator *MixLasso* for the estimation of generalized mixed regression, based on an atomic norm specifically constructed to regularize the number of mixture components. Our algorithm gives a risk bound that trades off between prediction accuracy and model sparsity without imposing stringent assumptions on the input/output distribution, and can be easily adapted to the case of non-linear functions. In our numerical experiments on mixtures of linear as well as nonlinear regressions, the proposed method yields high-quality solutions in a wider range of settings than existing approaches.

## 1   Introduction

The *Mixed Regression (MR)* problem considers the estimation of $K$ functions from a collection of input-output samples, where for each sample, the output is generated by one of the $K$ regression functions. When fitting linear functions in a noiseless setting, this is equivalent to solving $K$ linear systems, while at the same time, identifying which system each equation belongs to. The MR formulation can be employed as an approach to decompose a complicated function into $K$ simpler ones, by splitting the observations into $K$ classes. Variants of regression families such as *piecewise-linear regression* can be viewed as special cases of MR.

However, the MR problem is NP-hard in general [1] due to the simultaneous fitting of the discrete class labels as well as the regression functions. Standard approaches to the mixture problem employ local search methods such as Expectation Maximization (EM) [2] and Variational Bayes [3] that are prone to spurious local optima. There have thus been several lines of recent work studying estimation of mixed regression models with strong statistical guarantees under additional statistical assumptions. For the special case of linear function with $K=2$ components, [4] propose a convex nuclear norm minimization formulation that is guaranteed to estimate the two functions with minimax-optimal rates when given a sub-Gaussian design matrix. With the additional conditions of zero noise and isotropic Gaussian inputs, [1] propose an initialization for the EM algorithm to guarantee exact recovery of the true parameters. However, in addition to the stringent statistical assumptions, these methods and results are specialized to the case of two components, and seem non-trivial to generalize.

For problems with more than two components, most of the existing approaches [5, 6, 7, 8] rely on the *Tensor Methods*. In particular, for a $D$-dimensional linear MR problem, [6] propose a convex optimization formulation using a third-order tensor, which results in a computational cost of $O(ND^{12})$ and a sample complexity of $O(D^6/\epsilon^2)$, limiting its application to problems of small dimension. The *Tensor Decomposition* approach proposed in [5] has a sample complexity of only $O(D^3K^4/\epsilon^2)$ and is computationally efficient. However, it requires the knowledge of the input probability distribution in order to derive the *score function* used in their algorithm, which might not be available, and estimating the density over the $D$-dimensional input variables could be an even harder problem than MR itself. Other recent work [7, 8] show that in the noiseless setting with isotropic Gaussian inputs, an Alternating Minimization algorithm initialized with the Tensor Method leads to exact recovery of the true parameters. These latter methods have sample complexities linear in $D$, but with $O(K^K), O(K^{10})$ dependencies in $K$ respectively. Finally, [9] observed that, under the assumption of well-separated data, one can use a guaranteed clustering algorithm to find the mixture assignment of each observation, and thus solves the MR problem as a by-product. However, the data distribution considered in MR, such as those assumed in [5, 6, 7, 8], are usually not well-separated (see our Figure 3 as an example).

In this work, we address a generalized version of Mixed Regression where the output can be an additive combination of several mixture components. Our approach follows the general meta-approach emerging in the recent years of addressing latent-variable model estimation from the perspective of high-dimensional sparse estimation [10, 11, 12]. We propose a novel convex estimator *MixLasso* for the mixed regression problem, which enforces the mixture structure through minimizing a carefully constructed atomic norm that acts as a surrogate function for the number of mixture components. We then propose a greedy algorithm that generates a steepest-descent component at each iteration through solving a sub-problem similar to MAX-CUT. Our analysis of the algorithm gives a risk bound that trades off prediction accuracy and model sparsity, with a sample complexity that is linear in both $D$ and $K$, and without imposing any stringent assumptions on, or assuming knowledge of, the input/output distribution beyond that of boundedness, and even allowing for model mis-specification. This makes our *MixLasso* algorithm a theoretically sound method for a wide range of practical settings. Moreover, we also show how our proposed method can be easily extended to the nonlinear regression setting, to regression functions lying in a Reproducing Kernel Hilbert Space (RKHS). Our experiments with both generalized MR and standard MR show that the proposed method finds high-quality solutions in a wider range of settings when compared to existing approaches.

## 2   Generalized Mixed Regression

In Generalized Mixed Regression, the response $y \in \mathbb{R}$, given covariates $\boldsymbol{x} \in \mathcal{X}$, is specified as:

$$y = \sum_{k=1}^{K} z_k f_k(\boldsymbol{x}) + \omega \tag{1}$$

where $z_k \in \{0, 1\}$, $k = 1, \ldots, K$ is a latent binary vector indicating the presence or absence of each component, and $f_k(\boldsymbol{x}_i) : \mathbb{R}^D \to \mathbb{R}$ is the regression function of $k$-th component. The standard mixed regression is a special case of (1) with additional constraint $\|\boldsymbol{z}\|_0 = 1$. Here $\omega \in \mathbb{R}$ is a noise term with both bias and variance. In other words, we consider the very general setting where we allow for model mis-specification, and in general $\mathbb{E}[\omega | \boldsymbol{x}, \boldsymbol{z}] \neq 0$. This makes our problem setting in (1) very practically plausible, especially when the regression functions $\{f_k(\boldsymbol{x})\}_{k=1}^K$ lie in some restricted family such as linear functions. Our goal is to find $\mathcal{F} := \{f_k(\boldsymbol{x})\}_{k=1}^K$ minimizing the risk

$$r(\mathcal{F}) := \mathbb{E}\left[ \min_{\boldsymbol{z} \in \{0,1\}^K} \frac{1}{2}(y - \sum_{k=1}^{K} z_k f_k(\boldsymbol{x}))^2 \right], \tag{2}$$

while keeping the number of components $K$ as small as possible. This yields a trade-off between $r(\mathcal{F})$ and $K$. While one can always have a small risk with $K \to \infty$, we would like to find the smallest $K$ that achieves such risk.

## 3   MixLasso: Convex Estimation via Atomic Norm

In the following, we will first focus on the linear case $f_k(\boldsymbol{x}) := \langle \boldsymbol{w}_k, \boldsymbol{x} \rangle$ and consider extension to nonlinear functions in Section 4.2. Given a collection of i.i.d. samples $\{(\boldsymbol{x}_i, y_i)\}_{i=1}^N$, the $\ell_2$-

regularized Empirical Risk Minimization (ERM) problem for our task (2) is

$$\min_{W\in\mathbb{R}^{K\times D},\boldsymbol{z}_i\in\{0,1\}^K} \frac{1}{2N}\sum_{i=1}^{N}(y_i - \boldsymbol{z}_i^{\mathsf{T}}W\boldsymbol{x}_i)^2 + \frac{\tau}{2}\|W\|_F^2. \tag{3}$$

(3) is a hard optimization problem in general due to the simultaneous minimization w.r.t. parameters $W$ and binary hidden variables $\{\boldsymbol{z}_i\}_{i=1}^{N}$ [1]. However, given hidden variables, the problem is convex w.r.t. $W$, and thus, from the duality theory (3) is equivalent to

$$\min_{Z\in\{0,1\}^{N\times K}} \max_{\boldsymbol{\alpha}\in\mathbb{R}^N} \frac{-1}{N}\sum_{i=1}^{N}L^*(y_i, -\alpha_i) - \frac{1}{2N^2\tau}tr(\mathcal{D}(\boldsymbol{\alpha})XX^{\mathsf{T}}\mathcal{D}(\boldsymbol{\alpha})ZZ^{\mathsf{T}}) \tag{4}$$

where $Z := (\boldsymbol{z}_i)_{i=1}^{N}$, $\mathcal{D}(\boldsymbol{\alpha})$ is a diagonal matrix formed by vector $\boldsymbol{\alpha}$, and $L^*(y,\alpha) = y^{\mathsf{T}}\alpha + \frac{1}{2}\|\alpha\|^2$ is the convex conjugate of square loss $L(y,\xi) = \frac{1}{2}(y-\xi)^2$. The maximizer $\boldsymbol{\alpha}^*$ of (4) and minimizer $W^*$ of (3) are related by $W^* = \frac{1}{N\tau}\sum_{i=1}^{N}\alpha_i^*(\boldsymbol{z}_i\boldsymbol{x}_i^{\mathsf{T}}) = \frac{1}{N\tau}Z^{\mathsf{T}}D(\boldsymbol{\alpha}^*)X$.

A key observation for our formulation is that, although (4) is non-convex w.r.t. $Z$, it is a convex function of $M := ZZ^{\mathsf{T}}$ (since it is a maximum over linear functions of $M$). Therefore, the intractability of (4) only lies in the combinatorial constraint $M = ZZ^{\mathsf{T}}$ for some $Z \in \{0,1\}^{N\times K}$. To relax such constraint, we introduce an *atomic norm* [13] of the form

$$\|M\|_{\mathcal{S}} := \min_{c\geq 0} \sum_{a\in\mathcal{S}} c_a \ \ s.t. \ \ M = \sum_{a\in\mathcal{S}} c_a a. \tag{5}$$

where $\mathcal{S} := \{\boldsymbol{z}\boldsymbol{z}^{\mathsf{T}}|\boldsymbol{z} \in \{0,1\}^N\}$. Note if $c_a$ takes integer values $\{0,1\}$, $M = \sum_{a\in\mathcal{S}} c_a a = ZZ^{\mathsf{T}}$ for some $Z \in \{0,1\}^{N\times K}$ and $\|M\|_{\mathcal{S}} = K$. When $c_a$ is allowed to be any nonnegative number, (5) serves as a convex approximation to the number of components $K$ in a sense similar to $\ell_1$-norm as a convex approximation for the number of non-zero elements in *Lasso* [14]. Then the *MixLasso* estimator minimizes $\min_{M\in\mathbb{R}_+^{N\times N}} g(M) + \lambda\|M\|_{\mathcal{S}}$ where

$$g(M) := \max_{\boldsymbol{\alpha}\in\mathbb{R}^N} -\frac{1}{2N^2\tau}tr(\mathcal{D}(\boldsymbol{\alpha})XX^{\mathsf{T}}\mathcal{D}(\boldsymbol{\alpha})M) - \frac{1}{N}\sum_{i=1}^{N}L^*(y_i, -\alpha_i) \tag{6}$$

## 4 Algorithm

The convex formulation (6) is still a challenging optimization problem since it involves an atomic norm defined over $\bar{K} := 2^N$ atoms. An equivalent formulation expresses (6) as the minimizatioin of

$$F(\boldsymbol{c}) := g\bigg(\sum_{k=1}^{\bar{K}} c_k \boldsymbol{z}^k \boldsymbol{z}^{k\mathsf{T}}\bigg) + \lambda\|\boldsymbol{c}\|_1 \tag{7}$$

w.r.t. $\boldsymbol{c} \in \mathbb{R}_+^{\bar{K}}$, where $\{\boldsymbol{z}^k\}_{k=1}^{\bar{K}}$ enumerates $\forall \boldsymbol{z} \in \{0,1\}^N$. We introduce a greedy algorithm (Algorithm 1) for MixLasso, which maintains a sparse set of active components and adds one more active component $\boldsymbol{z}^k \boldsymbol{z}^{kT}$ at each iteration corresponding to the steepest descent direction

$$\min_{\boldsymbol{z}\in\{0,1\}^N} \langle \nabla g(M), \boldsymbol{z}\boldsymbol{z}^{\mathsf{T}}\rangle = -\frac{1}{2N^2\tau} \max_{\boldsymbol{z}\in\{0,1\}^N} \langle \mathcal{D}(\boldsymbol{\alpha}^*)XX^{\mathsf{T}}\mathcal{D}(\boldsymbol{\alpha}^*), \boldsymbol{z}\boldsymbol{z}^{\mathsf{T}}\rangle, \tag{8}$$

where $\boldsymbol{\alpha}^*$ is the maximizer in (6). As we show in Section 4.1, (8) is equivalent to a *MAX-CUT* like problem that can be solved efficiently with a constant-ratio approximation guarantee. Then we minimize (7) w.r.t. coefficients corresponding to the active components through a sequence of proximal gradient updates:

$$c_k^{s+1} \leftarrow \bigg[c_k^s - \frac{1}{\gamma|\mathcal{A}|}(\boldsymbol{z}^{k\mathsf{T}}\nabla g(M^s)\boldsymbol{z}^k + \lambda)\bigg]_+ \tag{9}$$

for $k \in \mathcal{A}$, and $s = 1\ldots S$, where $\gamma$ is the Lipschitz-continuous parameter of the coordinate-wise gradient $\boldsymbol{z}^{k\mathsf{T}}\nabla g(M)\boldsymbol{z}^k$. The evaluation of $\nabla g(M^s)$ involves finding the maximizer $\boldsymbol{\alpha}^*$, which can be obtained by solving the least-square problem:

$$W^* := \underset{W\in\mathbb{R}^{|\mathcal{A}|\times D}}{argmin} \frac{1}{2N}\sum_{i=1}^{N}(y_i - \boldsymbol{z}_i^{\mathsf{T}}W\boldsymbol{x}_i)^2 + \frac{\tau}{2}tr(W^{\mathsf{T}}\mathcal{D}^{-1}(\boldsymbol{c}_{\mathcal{A}})W) \tag{10}$$

---

**Algorithm 1** A Greedy Algorithm for MixLasso (6)

---

Initialize $\mathcal{A} = \emptyset$, $\boldsymbol{c} = 0$.
**for** $t = 1...T$ **do**
    1. Find a greedy component $\boldsymbol{z}\boldsymbol{z}^{\mathsf{T}}$ by solving (8).
    2. Add $\boldsymbol{z}\boldsymbol{z}^{\mathsf{T}}$ to the active set $\mathcal{A}$.
    3. Minimize (7) w.r.t. coordinates $\boldsymbol{c}_{\mathcal{A}}$ in the active set $\mathcal{A}$ through updates (9).
    4. Eliminate $\{\boldsymbol{z}^k \boldsymbol{z}^{k\mathsf{T}} | c_k = 0\}$ from $\mathcal{A}$.
**end for**.

---

and compute $\alpha_i^* = (y_i - \boldsymbol{z}_i^{\mathsf{T}} W^* \boldsymbol{x}_i)$. Let $E$ be the $N \times (|\mathcal{A}|D)$ design matrix of the least-square problem (10). By maintaining $E$, $E^{\mathsf{T}} E$ whenever the active set $\mathcal{A}$ changes, solving the least-square problem (10) costs $O(D^3 |\mathcal{A}|^3)$ amortizedly.

### 4.1 Greedy Generation of Components

Problem (8) for finding the steepest descent direction is a convex maximization problem with binary-valued variables and is hard in general. However, we show that it is equivalent to a *Boolean Quadratic Maximization* problem similar to *MAX-CUT*, where constant-ratio approximate algorithm exists through a Semidefinite Relaxation [15]. Furthermore, the Semidefinie Relaxation of this type has scalable solver that requires only complexity linear to the coefficient matrix [16, 17].

Let $C = \mathcal{D}(\boldsymbol{\alpha}^*) X X^{\mathsf{T}} \mathcal{D}(\boldsymbol{\alpha}^*)$. The greedy step (8) solves a problem of the form $\max_{\boldsymbol{z} \in \{0,1\}^N} \langle C, \boldsymbol{z}\boldsymbol{z}^{\mathsf{T}} \rangle$, which can be reduced to a problem of binary variables $\boldsymbol{v} \in \{-1, 1\}^N$ via a transformation $\boldsymbol{v} = 2\boldsymbol{z} - 1$:

$$\max_{\boldsymbol{v} \in \{-1,1\}^N} \frac{1}{4} \left( \langle C, \boldsymbol{v}\boldsymbol{v}^{\mathsf{T}} \rangle + 2\langle C, \mathbf{1}\boldsymbol{v}^{\mathsf{T}} \rangle + \langle C, \mathbf{1}\mathbf{1}^{\mathsf{T}} \rangle \right). \tag{11}$$

where $\mathbf{1}$ denotes $N$-dimensional vector of all 1s. By introducing a dummy variable $v_0$, (11) is equivalent to

$$\max_{(v_0; \boldsymbol{v}) \in \{-1,1\}^{N+1}} \frac{1}{4} \begin{bmatrix} v_0 \\ \boldsymbol{v} \end{bmatrix}^T \begin{bmatrix} \mathbf{1}^T C \mathbf{1} & \mathbf{1}^T C \\ C \mathbf{1} & C \end{bmatrix} \begin{bmatrix} v_0 \\ \boldsymbol{v} \end{bmatrix}. \tag{12}$$

Note one can always find a solution of $v_0 = 1$ by flipping signs of the solution since this does not change the objective value. Let the matrix in (12) be $\hat{C}$. Problem of form (12) is a Boolean Quadratic problem similar to MAX-CUT, for which there is Semidefinite relaxation of the form

$$\max_{V \in \mathbb{S}^N} \quad \langle \hat{C}, V \rangle$$
$$s.t. \quad V \succeq 0, \ diag(V) = \mathbf{1} \tag{13}$$

and rounding from which guarantees a solution $\hat{\boldsymbol{v}}$ to (12) satisfying $\overline{h} - h(\hat{\boldsymbol{v}}) \leq \rho(\overline{h} - \underline{h})$ with $\rho = 2/5$ [15], where $h(\boldsymbol{v})$ denotes the objective function of (12) and $\overline{h}, \underline{h}$ denote the maximum and minimum of the objective in (12) respectively. Note this result holds for any symmetric matrix $\hat{C}$. Since our problem has a positive-semidefinite matrix $\hat{C}$, we have $\underline{h} = 0$ and therefore the component $\boldsymbol{z}^k$ found this way satisfies

$$-\boldsymbol{z}^{k\mathsf{T}} \nabla g(M) \boldsymbol{z}^k = h(\hat{\boldsymbol{v}}) \geq \mu \overline{h} = \mu \max_{\boldsymbol{z} \in \{0,1\}^N} -\boldsymbol{z}^{\mathsf{T}} \nabla g(M) \boldsymbol{z} \tag{14}$$

with $\mu = 1 - \rho = 3/5$. Semidefinite Programming of the form (13) allows specialized solver with iteration cost linear to the matrix size $nnz(\hat{C})$ [16, 17]. And it is worth mentioning that, since our matrix $\hat{C}$ has low-rank structure (8), our implementation of the SDP solver [17] can further reduce the complexity per iteration from $nnz(\hat{C})$ to $nnz(X)$.

### 4.2 Nonlinear Extension

A simple way to consider a nonlinear version of the MixLasso estimator is to consider each component $f_k(\boldsymbol{x})$ lying in a Reproducing Kernel Hilbert Space (RKHS) $\mathcal{H}$ with respect to some Mercer kernel

$\mathcal{K}(\cdot, \cdot)$. In this setting, given $\{z_i\}_{i=1}^N$, the minimizer $\{f_k^*\}_{k=1}^K$ of

$$\min_{f_k \in \mathcal{H}} \frac{1}{2N} \sum_{i=1}^N \left( y_i - \sum_{k=1}^K z_{ik} f_k(\boldsymbol{x}_i) \right)^2 + \frac{\tau}{2} \sum_{k=1}^K \|f_k\|_{\mathcal{H}}^2 \tag{15}$$

satisfies the condition of the *Representer Theorem* that ensures an expression of the form $f_k^*(\boldsymbol{x}) = \sum_{i=1}^N \alpha_i z_{ik} \mathcal{K}(\boldsymbol{x}_i, \boldsymbol{x})$, $k \in [K]$, for the minimizer, and results in a MixLasso estimator (6) with

$$g(M) := \max_{\boldsymbol{\alpha} \in \mathbb{R}^N} -\frac{1}{2N^2\tau} tr(\mathcal{D}(\boldsymbol{\alpha}) Q \mathcal{D}(\boldsymbol{\alpha}) M) - \frac{1}{N} \sum_{i=1}^N L^*(y_i, -\alpha_i) \tag{16}$$

where $Q : N \times N$ is the kernel matrix with $Q_{ij} = \mathcal{K}(\boldsymbol{x}_i, \boldsymbol{x}_j)$. Then Algorithm 1 can be applied with the only difference on the evaluation of gradient $\nabla g(M)$, which requires finding the maximizer $\boldsymbol{\alpha}^*$ of (16) by solving the following linear system: $(\frac{1}{N\tau} Q \circ M + I)\boldsymbol{\alpha} = \boldsymbol{y}$. where $\circ$ denotes the elementwise product.

### 4.3 Rounding Procedure for Generalized & Standard Mixed Rregression

While the atomic-norm regularization $\lambda\|M\|_{\mathcal{S}}$ is a good convex relaxation of the number of components, the number of non-zero components getting from estimator (6) cannot be precisely specified apriori by the hyper-parameter $\lambda$ directly. In practice, it is often useful to obtain a solution $\boldsymbol{c}$ with exactly $\|\boldsymbol{c}\|_0 = K$ non-zeros. This can be achieved by setting the $K$ coefficients of largest magnitude to 1 and all the other coefficients to 0. This results in a $N \times K$ matrix of hidden assignments $\hat{Z}$ as the output of Algorithm 1. Then, starting from $\hat{Z}$, we can perform a number of alternating minimization steps between model parameters $W$ (or $\{f_k\}_{k=1}^K$ in general) and hidden assignments $\{z_i\}_{i=1}^N$ until convergence, as in a standard EM algorithm (with MAP hard assignment on $z_i$).

While we have proposed a solution of the generalized version (1), in some applications, it might be of interest to solve the special case of standard mixed regression, where each observation belongs to exactly one mixture component. One approach to convert a generalized mixture solution with $K$ components to a standard mixture of $J$ components is to find the most frequent $J$ patterns $\mathfrak{z}_1, \mathfrak{z}_2, ..., \mathfrak{z}_J$ from the estimated hidden assignments $\{\hat{z}_i\}_{i=1}^N$, and then force each observation to choose their hidden assignments $\{z_i\}_{i=1}^N$ from the set $\{\mathfrak{z}_j\}_{j=1}^J$ instead of arbitrary 0-1 patterns $\{0,1\}^K$. This results in $J$ functions $\{\mathfrak{f}_j\}_{j=1}^J$ of the form $\mathfrak{f}_j(\boldsymbol{x}) = \sum_{k=1}^K \mathfrak{z}_{jk} f_k(\boldsymbol{x})$, $j \in [J]$, being actually used in the training observation, and thus gives a valid model $\{\mathfrak{f}_j\}_{j=1}^J$ of standard mixed regression with $J$ components. Then as noted previously, one can further refine this rounded solution through EM iterates of standard mixed regression, initialized with component functions $\{\mathfrak{f}_j\}_{j=1}^J$.

## 5 Analysis

### 5.1 Convergence Analysis

We assume $y$ and $\boldsymbol{x}$ are bounded such that $|y| \leq R_y$, $\|\boldsymbol{x}\|_2 \leq R_x$. And without loss of generality, we assume the data are scaled such that $R_y = R_x = 1$. Then the following theorem guarantees the rate of convergence for Algorithm 1 up to a certain precision determined by the approximation ratio given in (14).

**Theorem 1.** *Let $F(\boldsymbol{c})$ be the objective* (7). *The greedy algorithm (Algorithm 1) satisfies*

$$F(\boldsymbol{c}^T) - F(\boldsymbol{c}^*) \leq \frac{2\gamma\|\boldsymbol{c}^*\|_1^2}{\mu^2} \left( \frac{1}{T} \right). \tag{17}$$

*for any iterate $T$ satisfying $F(\boldsymbol{c}^T) - F(\boldsymbol{c}^*) \geq \frac{2(1-\mu)}{\mu}\lambda\|\boldsymbol{c}^*\|_1$, where $\boldsymbol{c}^*$ is any reference solution, $\mu = 3/5$ is the approximation ratio given by (14) and $\gamma$ is the Lipschitz-continuous constant of the coordinate-wise gradient $\boldsymbol{z}^{k\mathsf{T}} \nabla g(M) \boldsymbol{z}^k$, $\forall k \in [\bar{K}]$.*

Then the following lemma shows that, with the additional assumption that $F(\boldsymbol{c})$ is strongly convex over a restricted support set $\mathcal{A}^*$, one can get a bound in terms of the $\ell_0$-norm of the reference solution.

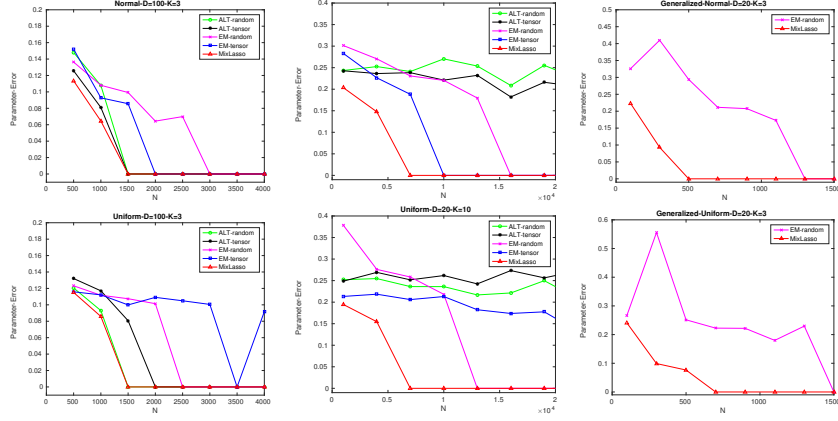

Figure 1: Results for Noiseless Mixture of Linear Regression with $N(0, I)$ input distribution (Top) and $U(-1, 1)$ input distribution (Bottom), where (Left) D=100, K=3, (Middle) D=20, K=10, and (Right) Generalized Mixture of Regression with D=20, K=3.

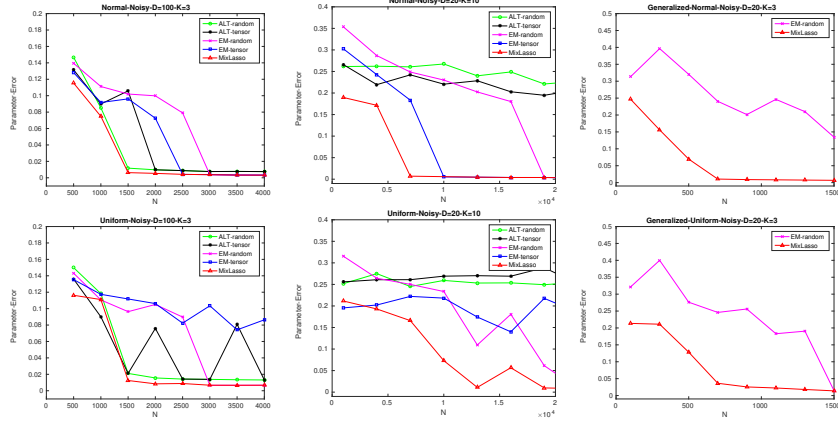

Figure 2: Results for Noisy ($\sigma = 0.1$) Mixture of Linear Regression with $N(0, I)$ input distribution (Top) and $U(-1, 1)$ input distribution (Bottom), where (Left) D=100, K=3, (Middle) D=20, K=10, and (Right) Generalized Mixture of Regression with D=20, K=3.

**Lemma 1.** *Let $\mathcal{A}^* \in [\bar{K}]$ be a support set and $\boldsymbol{c}^* := arg \min_{\boldsymbol{c}:supp(\boldsymbol{c})=\mathcal{A}^*} F(\boldsymbol{c}^*)$. Suppose $F(\boldsymbol{c})$ is strongly convex on $\mathcal{A}^*$ with parameter $\beta$. We have $\|\boldsymbol{c}^*\|_1 \leq \sqrt{\frac{2\|\boldsymbol{c}^*\|_0(F(0)-F(\boldsymbol{c}^*))}{\beta}}$.*

Since $F(0) - F(\boldsymbol{c}^*) \leq \frac{1}{2N} \sum_{i=1}^N y_i^2 \leq 1$, from (17), we have

$$F(\boldsymbol{c}^T) - F(\boldsymbol{c}^*) \leq \frac{4\gamma\|\boldsymbol{c}^*\|_0}{\beta\mu^2}\left(\frac{1}{T}\right) + \frac{2(1-\mu)\lambda}{\mu}\sqrt{\frac{2\|\boldsymbol{c}^*\|_0}{\beta}}. \tag{18}$$

for any $\boldsymbol{c}^* := arg \min_{\boldsymbol{c}:supp(\boldsymbol{c})=\mathcal{A}^*} F(\boldsymbol{c})$.

### 5.2 Generalization Analysis

In this section, we investigate the performance of output from Algorithm 1 in terms of the risk (2). Given a coefficients $\boldsymbol{c}$ with support $\mathcal{A}$, we can construct the weight matrix by $\hat{W}(\boldsymbol{c}) = \mathcal{D}(\sqrt{\boldsymbol{c}_{\mathcal{A}}})W$ with $W = Z_{\mathcal{A}}^\mathsf{T}\mathcal{D}(\boldsymbol{\alpha}^*)X$, where $Z_{\mathcal{A}} = (\boldsymbol{z}^k)_{k\in\mathcal{A}}$ and $\boldsymbol{\alpha}^*$ is the maximizer in (6) as a function of $\boldsymbol{c}$. From the duality between (3) and (4), $\hat{W}$ satisfies

$$F(\boldsymbol{c}_{\mathcal{A}}) = \frac{1}{2N} \sum_{i=1}^N (y_i - \boldsymbol{z}_i^\mathsf{T}\hat{W}\boldsymbol{x}_i)^2 + \frac{\tau}{2}\|W\|_F^2 + \lambda\|\boldsymbol{c}_{\mathcal{A}}\|_1. \tag{19}$$

The following theorem gives a risk bound for the output weight matrix $\hat{W}(\boldsymbol{c})$ from Algorithm 1.

**Theorem 2.** *Let $\mathcal{A}$, $\hat{\boldsymbol{c}}$, $\hat{W}$ be the set of active components, coefficients and corresponding weight matrix obtained from $T$ iterations of Algorithm 1, and $\bar{W}$ be the minimizer of the population risk (2) with $K$ components and $\|\bar{W}\|_F \leq R$. We have $r(\hat{W}) \leq r(\bar{W}) + \epsilon$ with probability $1 - \rho$ for $T \geq \frac{4\gamma}{\mu^2\beta}(\frac{K}{\epsilon})$ and $N = \Omega(\frac{DK}{\epsilon^3}\log(\frac{RK}{\epsilon\rho}))$ with $\lambda$, $\tau$ chosen appropriately as functions of $N$.*

Note the output of Algorithm 1 has number of components $\hat{K} \leq T$. Therefore, Theorem 2 gives a trade-off between the suboptimality of risk $r(\hat{W}) - r(\bar{W}) \leq \epsilon$ and number of components $\hat{K} = O(K/\epsilon)$. Note the result of Theorem (2) is obtained without distributional assumption on the input/output (except boundedness), so it is in general not possible to guarantee convergence to an optimal risk with exactly $K$ components, since finding such optimal solution is NP-hard even measured by the empirical risk [1]. It remains open if one can give a tighter result for the estimator (6) that achieves $\epsilon$-suboptimal risk with number of components being a constant multiple of $K$, or derive a bound on the parameter estimation error, possibly with additional assumptions on the observations.

# 6    Experiments

In this section, we compare the proposed *MixLasso* method with other state-of-the-art approaches listed as follows. (i) **EM-Random**: A standard EM algorithm that alternates between minimizing $\{\boldsymbol{z}_i\}_{i=1}^N$ and $\{f_k(\boldsymbol{x})\}_{k=1}^K$ until convergence, with random initialized $W \sim N(0, I)$ in the linear case and random initialized $Z \sim Multinoulli(1/K)$ in the nonlinear case. Each point in the figures is the best result out of 100 random trials. (ii) **EM-Tensor**: The EM algorithm initialized with Tensor Method proposed in [8]. The formula of Tensor Method is derived assuming $\boldsymbol{x}_i \sim N(0, I)$. We adopt implementation provided by the author of [7]. (iii) **ALT-Random**: An Alternating Minimization algorithm proposed in [7] with the same initialization strategy and number of trails as EM-Random. (iv) **ALT-Tensor**: The Alternating Minimization algorithm initialized with Tensor Method proposed in [7]. The formula of Tensor Method is derived assuming $\boldsymbol{x}_i \sim N(0, I)$. We adopt implementation provided by the author of [7]. (v) **MixLasso**: The proposed estimator with Algorithm 1. We round our solution to exact $K$ components according to the rounding procedure described in Section 4.3 for *generalized MR* and *standard MR* respectively. The rounded solution is further refined by EM iterates. For the linear case, we compare methods using the root mean square error on the learned parameters $W$ compared to the ground-truth parameters $W^*$ of size $K \times D$: $\min_{\mathcal{S}:|\mathcal{S}|=K} \frac{\|W_{\mathcal{S},:} - W^*\|_F}{\sqrt{DK}}$, where $\mathcal{S}$ denotes a multiset that selects the best matched row in $W$ for each row in $W^*$. For the nonlinear case, we compare methods using RMSE between the predicted value and the ground-truth function value: $\sqrt{\frac{1}{N}\sum_{i=1}^N(\sum_{k=1}^K z_{ik}f_k(\boldsymbol{x}_i) - \sum_{k=1}^K z_{ik}^* f_k^*(\boldsymbol{x}_i))^2}$.

## 6.1    Experiments on Synthetic Data

We generate 14 synthetic data sets according to the model: $y_i = \sum_{k=1}^K z_{ik}f_k(\boldsymbol{x}) + \omega_i,\ i \in [N]$, where *Syn1~Syn12* are generated by $D$-dimensional linear models $f_k(\boldsymbol{x}) = \boldsymbol{w}_k^T\boldsymbol{x}$ and *Syn13~Syn14* are generated by 1-dimensional polynomial model of degree 6: $f_k(x) = \sum_{j=1}^6 w_{kj}x^j$. Figure 1 and 2 give experimental results of the linear model in the noiseless and noisy case respectively. We observe that, in the case of Normal input distribution (Syn1, Syn2, Syn7, Syn8) (top row), both the Tensor-initialized methods and MixLasso consistently improve upon random-initialized EM/ALT (even with 100 trials) in terms of the number of samples required to achieve a good performance, where *ALT* performs better than *EM* in higher dimensional case ($D = 100, K = 3$) while *EM* performs better for cases of more components ($D = 20, K = 10$); meanwhile, MixLasso leads to significant improvements in both cases. On the other hand, when the input distribution becomes U(-1,1) (Syn4, Syn5, Syn10, Syn11), the tensor-initialized method becomes even worse than the random-initialized ones, presumably due to the model mis-specification, while MixLasso still consistently improve upon the random initialized EM/ALT. Note we are testing *Tensor Method derived based on the Normal assumption* on *data with Uniform input* on purpose. The fgoal is to see the effect of model misspecification on the Tensor approach, as in practice one would always have model misspecification to some degree. The rightmost columns of Figure 1, 2 show the results on data generated from the *generalized mixed regression* model (Syn3, Syn6, Syn9, Syn12), where Tensor-based methods

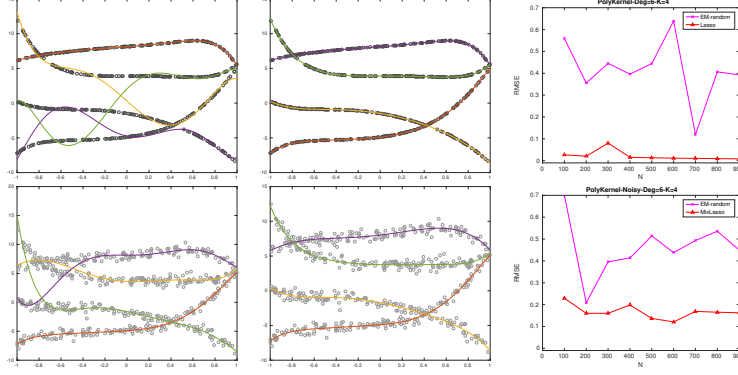

Figure 3: Results on Mixture of 6th-order Polynomial Regression of $K$=4 components with noise (Bottom) and without noise (Top). (Left) The best result of EM out of 100 random initialization. (Middle) Solution from MixLasso followed by fine-tuning EM iterates. (Right) Comparison in terms of RMSE.

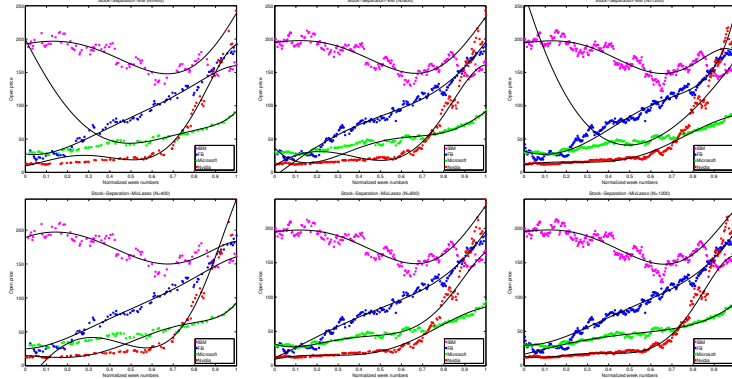

Figure 4: Results of fitting mixture of polynomial regressions on the *Stock* data set of increasing number of samples. The top row shows results fitted by EM, and the bottom row shows that from MixLasso. From left to right we have (left) 100 weeks, (middle) 200 weeks, and (right) 300 weeks. From left to right, the RMSE of **EM=**$(\mathbf{6.33}, \mathbf{6.04}, \mathbf{6.27})$ and the RMSE of **MixLasso=**$(\mathbf{6.29}, \mathbf{5.75}, \mathbf{5.58})$.

are not applicable, while MixLasso improves upon EM-Random by a large margin. Figure 3 gives a comparison of EM-Random and MixLasso on Mixture of Kernel Regression with polynomial kernel $\mathcal{K}(\boldsymbol{x}_i, \boldsymbol{x}_j) = (a\boldsymbol{x}_i^{\mathsf{T}}\boldsymbol{x}_j + b)^d$ $(d = 6)$, where we generate $K$=4 random 6th-degree polynomial functions $\{f_k^*\}_{k=1}^K$ by uniform sampling their coefficients from $U(-4, 4)$. In this setting, we found EM-Random has a hard time converging to the ground-truth solution even with 100-restarts, while MixLasso obtains solution close to the ground truth with a small number of samples.

## 6.2 Experiments on Real Data

In this section, we compare *MixLasso* and *EM* (with 100 restarts) for fitting a mixture of polynomial regressions on a *Stock* data set that contains the mixed stock prices of *IBM*, *Facebook*, *Microsoft* and *Nvidia* of span 300 weeks till the Feb. of 2018. The task is to automatically recover the company label of each stock price, while fitting the stock price time series of each company as a polynomial curve. Both EM and MixLasso use a polynomial kernel of the parameters: $\mathcal{K}(\boldsymbol{x}_i, \boldsymbol{x}_j) = (2\boldsymbol{x}_i^{\mathsf{T}}\boldsymbol{x}_j + 2)^8$. The results are shown in Figure 4. We can see that MixLasso almost recovers the pattern when all samples are given, except for a small number of samples generated by Nvidia's rapid growth recently. While MixLasso consistently achieving a lower RMSE over different sample sizes, the RMSE gap between MixLasso and EM increases as the number of samples grows.

**Acknowledgements.** P.R. acknowledges the support of NSF via IIS-1149803, IIS-1664720, DMS-1264033, and ONR via N000141812861.

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
