[Supplementary Material]

# 7 Appendix

## 7.1 Proof for Theorem 1

Let $g(M)$ be a smooth function such that $\nabla g(M)$ is Lipschitz-continuous with parameter $\rho$, that is,

$$g(M') - g(M) - \langle \nabla g(M), M' - M \rangle \leq \frac{\rho}{2} \|M' - M\|_F^2.$$

Then $\nabla_j f(c) = z_j^T \nabla g(M) z_j$ is Lipschitz-continuous with parameter $\gamma$, a number of order $O(1)$ when $g(.)$ is an empirical risk normalized by $N$. Let $\mathcal{A}$ be the active set before adding a component $\hat{j}$. Consider the descent amount produced by minimizing $F(c)$ w.r.t. the $c_{\hat{j}}$ given that $0 \in \partial_j F(c)$ for all $j \in \mathcal{A}$ due to the subproblem in the previous iteration. Let $j = \hat{j}$, for any $\eta_j$ we have

$$F(c + \eta_j e_j) - F(c) \leq \nabla_j f(c) \eta_j + \lambda |\eta_j| + \frac{\gamma}{2} \eta_j^2$$

$$\leq \mu \nabla_{j^*} f(c) \eta_j + \lambda |\eta_j| + \frac{\gamma}{2} \eta_j^2$$

$$\leq \mu \nabla_{j^*} f(c) \eta_j + \lambda |\eta_j| + \frac{\gamma}{2} \eta_j^2$$

Minimize w.r.t $\eta_j$ gives

$$\min_{\eta_j} \ F(c + \eta_j e_j) - F(c)$$

$$\leq \min_{\eta_j} \ \mu \nabla_{j^*} f(c) \eta_j + \lambda |\eta_j| + \frac{\gamma}{2} \eta_j^2$$

$$\leq \min_{\eta_j} \ \mu \nabla_{j^*} f(c) \eta_j + \lambda |\eta_j| + \frac{\gamma}{2} \eta_j^2$$

$$= \min_{\eta_k : k \notin \mathcal{A}} \ \sum_{k \notin \mathcal{A}} \left( \mu \nabla_k f(c) \eta_k + \lambda |\eta_k| \right) + \frac{\gamma}{2} \left( \sum_{k \notin \mathcal{A}} |\eta_k| \right)^2$$

$$\leq \min_{\eta_k : k \notin \mathcal{A}} \ \mu \sum_{k \notin \mathcal{A}} \left( \nabla_k f(c) \eta_k + \lambda |\eta_k| \right) + \frac{\gamma}{2} \left( \sum_{k \notin \mathcal{A}} |\eta_k| \right)^2$$

$$+ (1 - \mu) \lambda \sum_{k \notin \mathcal{A}} |\eta_k|$$

where the last equality is justified by Lemma 2 provided later. For $k \in \mathcal{A}$, we have

$$0 = \min_{\eta_k : k \in \mathcal{A}} \ \mu \sum_{k \in \mathcal{A}} \left( \nabla_k f(c) \eta_k + \lambda |c_k + \eta_k| - \lambda |c_k| \right)$$

Combining cases for $k \notin \mathcal{A}$ and $k \in \mathcal{A}$, we can obtain a global estimate of descent amount compared to some optimal solution $x^*$ as follows

$$\min_{\eta_{\hat{j}}} \ F(c + \eta_{\hat{j}} e_{\hat{j}}) - F(c)$$

$$\leq \min_{\eta} \ \mu \left( \langle \nabla f(c), \eta \rangle + \lambda \|c + \eta\|_1 - \lambda \|c\|_1 \right)$$

$$+ \frac{\gamma}{2} \left( \sum_{k \notin \mathcal{A}} |\eta_k| \right)^2 + (1 - \mu) \lambda \sum_{k \notin \mathcal{A}} |\eta_k|$$

$$\leq \min_{\eta} \ \mu \left( F(c + \eta) - F(c) \right) + \frac{\gamma}{2} \left( \sum_{k \notin \mathcal{A}} |\eta_k| \right)^2 + (1 - \mu) \lambda \sum_{k \notin \mathcal{A}} |\eta_k|$$

$$\leq \min_{\alpha \in [0,1]} \ \mu \left( F(c + \alpha(c^* - c)) - F(c) \right) + \frac{\alpha \gamma}{2} \|c^*\|_1^2 + \alpha(1 - \mu) \lambda \|c^*\|_1$$

$$\leq \min_{\alpha \in [0,1]} \ -\alpha \mu \left( F(c) - F(c^*) \right) + \frac{\alpha^2 \gamma}{2} \|c^*\|_1^2 + \alpha(1 - \mu) \lambda \|c^*\|_1.$$

It means we can always choose an $\alpha$ small enough to guarantee descent if

$$F(c) - F(c^*) > \frac{(1-\mu)}{\mu} \lambda \|c^*\|_1. \tag{20}$$

Then for

$$F(c) - F(c^*) \geq \frac{2(1-\mu)}{\mu} \lambda \|c^*\|_1, \tag{21}$$

we have

$$\min_{\eta_{\hat{j}}} F(c + \eta_{\hat{j}} e_{\hat{j}}) - F(c)$$

$$\leq \min_{\alpha \in [0,1]} -\frac{\alpha\mu}{2}\left(F(c) - F(c^*)\right) + \frac{\alpha^2\gamma}{2}\|c^*\|_1^2.$$

Minimizing w.r.t. to $\alpha$ gives the convergence guarantee

$$F(c^t) - F(c^*) \leq \frac{2\gamma\|c^*\|_1^2}{\mu^2}\frac{1}{t}.$$

for any iterate with $F(c^t) - F(c^*) \geq \frac{2(1-\mu)}{\mu}\lambda\|c^*\|_1$.

**Lemma 2.**

$$\min_{\eta_j} \ \mu\nabla_{j^*}f(c)\eta_j + \lambda|\eta_j| + \frac{\gamma}{2}\eta_j^2 \tag{22}$$

$$= \min_{\eta_k: k \notin \mathcal{A}} \ \sum_{k \notin \mathcal{A}} \left(\mu\nabla_k f(c)\eta_k + \lambda|\eta_k|\right) + \frac{\gamma}{2}\left(\sum_{k \notin \mathcal{A}}|\eta_k|\right)^2 \tag{23}$$

*Proof.* The minimization (28) is equivalent to

$$\min_{\eta_k: k \notin \mathcal{A}} \ \sum_{k \notin \mathcal{A}}\left(\mu\nabla_k f(c)\eta_k\right)$$

$$s.t. \quad \left(\sum_{k \notin \mathcal{A}}|\eta_k|\right)^2 \leq C_1 \ , \ \sum_{k \notin \mathcal{A}}|\eta_k| \leq C_2.$$

and therefore is equivalent to

$$\min_{\eta_k: k \notin \mathcal{A}} \ \mu\sum_{k \notin \mathcal{A}}\nabla_k f(c)\eta_k$$

$$s.t. \quad \sum_{k \notin \mathcal{A}}|\eta_k| \leq \min\{\sqrt{C_1}, C_2\}$$

which is a linear objective subject to a convex set and thus always has solution that lies on the corner point with only one non-zero coordinate $\eta_{j^*}$, which then gives the same minimum as (27). □

## 7.2 Proof for Lemma 1

Since $supp(c^*) = \mathcal{A}^*$, and $c^*$ is optimal when restricted on the support, we have $\langle \eta, c^* \rangle = 0$ for some $\eta \in \partial F(c^*)$. And since $F(c)$ is strongly convex on the support $\mathcal{A}^*$ with parameter $\beta$, we have

$$F(0) - F(c^*) = F(0) - F(c^*) - \langle \eta, 0 - c^* \rangle$$

$$\geq \frac{\beta}{2}\|c^* - 0\|_2^2,$$

which gives us

$$\|c^*\|_2^2 \leq \frac{2(F(0) - F(c^*))}{\beta}.$$

Combining above with the fact for any $c$, $\|c\|_1^2 \leq \|c\|_0\|c\|_2^2$, we obtain the result.

## 7.3 Proof for Theorem 2

**Lemma 3.** *Let $r(W)$ and $r_N(W)$ be the risk (2) and the empirical risk respectively, we have*

$$\sup_{W \in \mathbb{R}^{K \times D}: \|W\|_F \leq R} |r(W) - r_N(W)|$$

$$\leq \sqrt{\frac{2DK \log(4RKN)}{N} + \frac{1}{N}\log(\frac{1}{\rho})}$$

*with probability $1 - \rho$.*

*Proof.* Since $\min_{\boldsymbol{z} \in \{0,1\}^N} \frac{1}{2}(y - \boldsymbol{z}^\mathsf{T} W\boldsymbol{x})^2 \leq |y|^2 \leq 1$ for a given $W$, by Hoeffding inequality,

$$P\left(|r_N(W) - r(W)| \geq \epsilon\right)$$
$$\leq \exp(-2N\epsilon^2).$$

Let $\mathcal{N}(\delta)$ be a $\delta$-covering of the set $\mathcal{W} := \{W \in \mathbb{R}^{K \times D} \mid \|W\|_F \leq R\}$ with $|\mathcal{N}(\delta)| \leq \left(\frac{4R}{\delta}\right)^{DK}$. Then for any $W \in \mathcal{W}$, we have $\tilde{W} \in \mathcal{N}(\delta)$ with $\|W - \tilde{W}\| \leq \delta$. Applying a union bound, we have

$$P\left(\sup_{\tilde{W} \in \mathcal{N}(\delta)} |r_N(\tilde{W}) - r(\tilde{W})| \geq \epsilon\right)$$
$$\leq \left(\frac{4R}{\delta}\right)^{DK} \exp(-2N\epsilon^2). \tag{24}$$

Then for $\Delta W := W - \tilde{W}$ satisfying $\|\Delta W\| \leq \delta$, we can bound the difference of square loss of $W$ and $\tilde{W}$ by

$$\min_{\boldsymbol{z} \in \{0,1\}^K} \frac{1}{2}(y - \boldsymbol{z}^\mathsf{T} W\boldsymbol{x})^2 - \min_{\boldsymbol{z} \in \{0,1\}^K} \frac{1}{2}(y - \boldsymbol{z}^\mathsf{T} \tilde{W}\boldsymbol{x})^2$$
$$\leq \frac{1}{2}(y - \tilde{\boldsymbol{z}}^\mathsf{T} W\boldsymbol{x})^2 - \frac{1}{2}(y - \tilde{\boldsymbol{z}}^\mathsf{T} \tilde{W}\boldsymbol{x})^2 \tag{25}$$
$$\leq \|\Delta W\|_F \|\tilde{\boldsymbol{z}}\| + 2R\|\tilde{\boldsymbol{z}}\|^2 \|\Delta W\|_F \leq 3RK\epsilon$$

where $\tilde{\boldsymbol{z}} = arg\min_{\boldsymbol{z} \in \{0,1\}^K} \frac{1}{2}(y - \boldsymbol{z}^\mathsf{T} W\boldsymbol{x})^2$ and we used the fact that $\|\boldsymbol{x}\| \leq 1$ and $|y| \leq 1$. By symmetry, we have

$$\left| \min_{\boldsymbol{z} \in \{0,1\}^K} \frac{1}{2}(y - \boldsymbol{z}^\mathsf{T} \tilde{W}\boldsymbol{x})^2 - \min_{\boldsymbol{z} \in \{0,1\}^K} \frac{1}{2}(y - \boldsymbol{z}^\mathsf{T} W\boldsymbol{x})^2 \right| \leq 3RK\epsilon$$

. Combining (24) with (25), we have

$$\sup_{W \in \mathcal{W}} |r_N(W) - r_N(W)|$$
$$\leq 6RK\delta + \sqrt{\frac{DK}{2N}\log(\frac{4R}{\delta}) + \frac{1}{2N}\log(\frac{1}{\rho})}. \tag{26}$$

with probability $1 - \rho$. Setting $\delta = 1/(6RK\sqrt{N})$ and apply Jennen's inequality gives the result. □

Then the following gives the proof for Theorem 2.

*Proof.* Let $\bar{\boldsymbol{z}}_i = arg\min_{\boldsymbol{z}_i \in \{0,1\}^K} (y_i - \boldsymbol{z}_i^\mathsf{T} \bar{W}\boldsymbol{x}_i)^2$ for $i \in [N]$. Denote $\bar{Z}$ as the $N \times K$ matrix stacked from $(\bar{\boldsymbol{z}}_i^\mathsf{T})_{i=1}^N$. Let $\{\bar{\boldsymbol{z}}^k\}_{k=1}^K$ be the columns of $\bar{Z}$ and $\bar{A}$ be the indexes of atoms in the atomic set (5) that have the same 0-1 patterns to those columns. Denote $\bar{\boldsymbol{c}}$ as the coefficient vector with $\bar{c}_k = 1$ for $k \in \bar{A}$ and $\bar{c}_k = 0$ for $k \notin \bar{A}$. By the definition of $F(\boldsymbol{c})$, we have

$$F(\bar{\boldsymbol{c}}) \leq r_N(\bar{W}) + \frac{\tau}{2}\|\bar{W}\|_F^2 + \lambda\|\bar{\boldsymbol{c}}\|_1. \tag{27}$$

where $r_N(\bar{W}) := \frac{1}{2N} \sum_{i=1}^{N} \min_{\mathbf{z} \in \{0,1\}^K} (y_i - \mathbf{z}^\mathsf{T} \bar{W} \mathbf{x}_i)^2$ is the empirical risk of $\bar{W}$. Let $\mathbf{c}^* := arg \min_{\mathbf{c}:supp(\mathbf{c})=\bar{A}} F(\mathbf{c})$. We have $F(\mathbf{c}^*) \le F(\bar{\mathbf{c}})$. Then from (18),

$$F(\hat{\mathbf{c}}) - F(\bar{\mathbf{c}}) \le F(\hat{\mathbf{c}}) - F(\mathbf{c}^*) \le \frac{4\gamma K}{\beta\mu^2} \left( \frac{1}{T} \right) + \frac{2(1-\mu)\lambda}{\mu} \sqrt{\frac{2K}{\beta}}. \tag{28}$$

In addition, the risk of $\hat{W}$ satisfies

$$r_N(\hat{W}) + \frac{\tau}{2} \|\hat{W}\|_F^2 + \lambda \|\hat{\mathbf{c}}\|_1 \le F(\hat{\mathbf{c}}) \tag{29}$$

by the definition of the empirical risk $r_N(.)$ (since it is minimized w.r.t. the hidden assignments). Combining (27), (28) and (29), we obtain a bound on the difference of empirical risk

$$r_N(\hat{W}) - r_N(\bar{W})$$

$$\le \underbrace{\frac{\tau}{2} \|\bar{W}\|^2 + \lambda K}_{\text{bias of regularization}} + \underbrace{\frac{4\gamma K}{\beta\mu^2} \left( \frac{1}{T} \right) + \frac{2(1-\mu)\lambda}{\mu} \sqrt{\frac{2K}{\beta}}}_{\text{optimization error}} \tag{30}$$

The remaining task is to bound the estimation error $|r(W) - r_N(W)|$. Since Algorithm 1 is a descent algorithm w.r.t. $F(\mathbf{c})$ and in the beginning $F(0) \le 1/2$, we have $\|\mathbf{c}\|_1 \le 1/\lambda$ and $\|W\|^2 \le 1/\tau$ at any iterate. Then we can bound the estimation error by Lemma 3 for $\hat{W}$ belonging to the set $\mathcal{W}(T) := \{\hat{W} \in \mathbb{R}^{T \times D} \mid \|\hat{W}\|_F \le \sqrt{1/\lambda\tau}\}$, giving

$$|r(\hat{W}) - r_N(\hat{W})| \le \sqrt{\frac{2DT \log(4TN/\sqrt{\lambda\tau})}{N} + \frac{1}{N} \log(\frac{1}{\rho})}. \tag{31}$$

Combining (30) and (31), and choosing $\lambda = 1/(NK)$, $\tau = 1/(NR^2)$, we obtain $r(\hat{W}) - r(\bar{W}) \le \epsilon$ with $T \ge \frac{4\gamma}{\mu^2\beta}(\frac{K}{\epsilon})$, and $N \ge \frac{DT}{\epsilon^2}(2\log(\frac{4RKT}{\epsilon}) + \log(\frac{1}{\rho}))$ for any $0 < \epsilon < 1$. $\square$