[Reviews · NeurIPS 2018]

Reviewer 1



This paper describes an approximate approach to learning a generalized mixed regression. First, the problem is approximated by empirical risk minimization. Then, a combinatorial constraint is relaxed to an atomic norm penalty. The algorithm proposed to solve the derived problem is a greedy one, where at each iteration a MAX-CUT-like subproblem is approximately solved (up to a constant-ratio approximation) by a semidefinite relaxation (similar to the one commonly associated with MAX-CUT). The paper includes a convergence analysis covering the algorithm, and a generalization analysis covering the original approximation/relaxation of the original problem formulation. The result is relevant to a core tool in machine learning (mixed regression). The writing is remarkably clear and the exposition easy to follow. The end of Section 5 reconciles the gap between the guarantees of this paper and the difficulty of solving original problem, which is NP-hard. The experiments are thoroughly described and designed in a way that makes natural comparisons and illustrates the observed behavior of the algorithm. One point of feedback, which I see as optional, but that may benefit the presentation, would be to motivate the generalized mixed regression problem by relating it to other problems. Are there any interesting special cases of generalized mixed regression other than mixed regression? Are other generalizations possible, and are they harder or easier? Typo on line 119: "semidefinie" -> semidefinite Line 182: formatting issue with "argmin" expression.

Reviewer 2



This paper considers a generalized version of the mixed regression problem, where we observe a collection of input-output samples, with the output corresponding to an additive combination of several mixture components/functions, and the goal is to find a collection of K functions that minimise the risk. The corresponding ERM problem is NP-Hard to solve due to combinatorial constraints. The authors propose to relax these constraints by replacing them with an atomic norm regularizer they introduce as an "approximation" of the number of components. They propose to solve the resulting convex problem using a greedy algorithm. Their analysis show that the solutions obtained by their approach achieve epsilon-optimal risk using a linear number of samples (both in terms of K and the dimension D) and O(K/epsilon) number of components, thus improving over the state-of-the-art in terms of number of sample complexity. The numerical experiments confirm this improvement. The paper is written in a fairly clear way, and the approach of solving a convex relaxation of the mixed regression problem and the obtained sample complexity are compelling contributions. However, the introduced greedy algorithm (Algo. 1) seems unnecessarily computationally expensive! In particular, step 3 of Algorithm 1 requires minimizing the loss function (7) over the current active set. The authors propose to solve this using proximal gradient algorithm (eq. 9), which requires solving a least squares problem at each iteration. Moreover, the convergence rate provided in Theorem 1 assumes that step (3) is solved exactly, which is not possible. Note also that the obtained convergence rate does not guarantee eps-optimal solution: F(c^T) - F(c^*) <= max{ O( ||c^*||_1 ), O(||c^*||^2_1 / T)}. Why not solve instead the constrained formulation min g(M) s.t. ||M||_S <= K? This can simply be solved by Frank-Wolfe algorithm which only requires step 1 of Algorithm 1, and has a known convergence rate of O(1/T). I think the paper could be improved by replacing Algorithm 1 with Frank-Wolfe and modifying the analysis in Theorem 2 accordingly. Update: I thank the authors for their responses to my questions. Please note that Frank-Wolfe variants for the regularized form were also considered in the literature. For example, the paper "Conditional Gradient Algorithms for Norm-Regularized Smooth Convex Optimization" by Z. Harchaoui, A. Juditsky, and A. Nemirovski, provides a variant of Frank-Wolfe, which precisely address problems regularized with a norm, and is more computationally efficient than the proposed Algorithm 1. It is worth mentioning such algorithms in the paper, with an explanation (if any) on why Algorithm 1 might still be preferable in this case (for ex because it leads to easier analysis).

Reviewer 3



The paper proposes a novel algorithm and analysis for solving mixed regression problem. The paper uses atomic norm to relax the original problem which is intractable due to the combinatorial nature of determining which component the data come from. In solving each step of the greedy step, the algorithm takes advantage of a solution similar to MAX_CUT and arrives at an efficient optimization algorithm. The authors also back up the proposed algorithms with convergence and generalization analysis. Experiments on synthetic and real datasets demonstrate the solid performance compared to other state-of-the-arts algorithms. The paper is very clearly written and it is a pleasure to read. It is organized very nicely and the results are presented in a clear manner. In addition, the motivation, and the description of the algorithm, and the experiments are all clearly presented. More importantly, the algorithm is novel and wth good theoretical guarantees. In fact, the sample complexity is much better than tensor decomposition based approaches. It is interesting that convex relaxation achieves such good results on this problem. The experiments are also solid. The paper presents extensive results compared to other approaches, and the performance is significantly better. Though, it will be interesting to also compare time complexity to verify its scalability to large datasets and to large number of components. ============================================================ From other reviews and the rebuttal, it seems there is more efficient optimization algorithm in the literature, which is not discussed in the paper. The authors should add some discussion about this issue.